# Preparation of Organo-Montmorillonite Modified Poly(lactic acid) and Properties of Its Blends with Wood Flour

**DOI:** 10.3390/polym11020204

**Published:** 2019-01-24

**Authors:** Ru Liu, Xiaoqian Yin, Anmin Huang, Chen Wang, Erni Ma

**Affiliations:** 1Research Institute of Wood Industry, Chinese Academy of Forestry, Haidian, Beijing 100091, China; liuru@criwi.org.cn (R.L.); yxqwzy@163.com (X.Y.); wonderfulmorning@163.com (C.W.); 2MOE Key Laboratory of Wooden Material Science and Application, Beijing Forestry University, Qinghua East Road 35, Haidian, Beijing 100083, China

**Keywords:** Dispersion, exfoliated silicate layers, interfacial adhesion, montmorillonite, wood–plastic composites

## Abstract

In this study, poly(lactic acid) (PLA)/wood flour (WF) composites were prepared by first blending PLA with organo-montmorillonite (OMMT) at different contents (0.5, 1, 1.5, and 2 wt %). The physical and mechanical properties of the virgin and OMMT modified PLA and its WF composites were tested. The results showed that: (1) at low OMMT content (<1 wt %), OMMT can uniformly disperse into the PLA matrix with highly exfoliated structures. When the content increased to 1.5 wt %, some aggregations occurred; (2) after a second extruding process, the aggregated OMMT redistributed into PLA and part of OMMT even penetrated into the WF cell wall. However, at the highest OMMT content (2 wt %), aggregates still existed; (3) the highly exfoliated OMMT was beneficial to the physical and mechanical properties of PLA and the WF composites. The optimal group of OMMT-modified PLA was found at an OMMT content of 0.5 wt %, while for the PLA/WF system, the best properties were achieved at an OMMT content of 1.5 wt %.

## 1. Introduction

Sustainable and efficient resource utilization becomes more and more important due to the continuously growing demand for resources. As an alternative material for fossil fuels, wood–polymer composites (WPC) are a group of hybrid materials which are prepared from renewable waste wood fiber/flour (WF) and polymer. Compared with classic fibers, WF represents superior properties when they are used as reinforcing fillers in a polymer matrix, such as lost cost, non-toxicity, recyclability, high strength, and specific stiffness [1,2,3]. By blending WF with a hydrophobic polymer matrix, the moisture absorption and sensitivity to fungal decay of WF are reduced. Simultaneously, the WF enhances the stiffness, thermal stability, and restricts the creep behavior of the polymer [4]. Therefore, WPC products have gained a market share in various applications, such as decks, railings, fencing, constructions, furniture, and automotive components [5,6,7]. As a thermoplastic matrix material, polyvinyl chloride (PVC), polyethylene (PE), and polypropylene (PP) are commonly used for WPC [8]. Considering the decreasing fossil resources and increasing amounts of plastic waste, the market for biopolymers is rapidly growing, such as poly(lactic acid) (PLA). Similar to traditional WPC, the properties of PLA/WF composites also strongly depend on the compatibility and interfacial adhesion between WF and PLA, which represent one of the main limitations as wood is strongly polar and PLA is nonpolar [9]. To overcome this problem, chemical modification of WF is applied or additives like coupling agents are used [10,11,12,13,14].

Using nanoparticles as fillers in polymer composites has been achieved with improvements in stiffness, strength, and the glass transition temperature of the polymers due to the large interfaces developed with the dispersion of nanoparticles in the polymer matrix [15,16,17]. The large surface area influences the molecular bonding of the polymer crosslinking, and in turn, enhances the mechanical strength of the polymers. Montmorillonite (MMT) is a kind of nanoparticle which attracts great attention in academic and industrial sectors for its abundance and high performance with regard to the aspect ratio and specific surface area [18,19,20]. Therefore, adding a small amount of MMT can substantially enhance the physical and mechanical properties of polymers. Natural MMT contains many inorganic cations in its gaps, such as Na^+^ and Ca^2+^, showing a high polarity and difficulty of dispersion in non-polar polymers [21]. Introducing the alkylammonium cations into MMT interlayers and replacing the small exchangeable cations to form organo-montmorillonite (OMMT) is the most common method used [22].

In most studies, the introduction of OMMT into WPC was carried out by simple ternary blending with WF and polymers [23,24,25]. This process is convenient, but the improvements are limited because OMMT is easily aggregated and attached to the WF surface, which causes weak interfacial layers [26]. Meng et al. [27] found that better performance of the OMMT/PLA/WF composite was achieved in a two-step method where OMMT-modified PLA matrices were first prepared by compounding OMMT with virgin PLA. However, some flocculated OMMT were still observed, and the mechanical strength decreased compared with virgin PLA. In a previous study, an organo-MMT (OMMT) was successfully prepared with sodium-MMT (Na-MMT) and didecyl dimethyl ammonium chloride (DDAC) within the wood fiber [28]. The physical and mechanical properties of the resulting PLA/WF composites were significantly improved, which may be owed to the double-chained quaternary ammonium structure of DDAC [29]. Therefore, DDAC could be used as a good modifier to make composites with a low environmental impact. In this study, the DDAC-based OMMT was first prepared and then blended with virgin PLA through extrusion at different OMMT contents (0.5, 1, 1.5, and 2 wt %). The thus-modified PLA was blended with WF by extrusion followed with hot-pressing. Some physical and mechanical properties of the composites were studied. The composites were characterized by X-ray diffraction (XRD), scanning electron microscopy (SEM), and transmission electron microscope (TEM) analyses.

## 2. Materials and Methods

### 2.1. Materials

The wood flour of poplar (*Populus tomentasa* Carr.) with mesh size of 10–60 was kindly donated by Xingda Wood Flour Company, Gaocheng, China. It has an average length of 1.5 mm and an average diameter of 0.2 mm. PLA (AI-1001) (L content) was obtained from Esun, Shenzhen BrightChina Industrial Co., Ltd, Guangzhou, China. It has a density of around 1.25 g/cm^3^ and a melt flow index of about 12–15 g/10min at 190 °C with a load of 2.16 kg. The *M*_n_ of PLA is about 80,000 g/mol. Na-MMT (PGV; Nanocor Inc, Chicago, IL, USA) was purchased from East West Company, Beijing, China. It is a hydrophilic clay powder with a specific gravity of 2.6 and its pH at 5% *w*/*w* in water is 9–10. The cation exchange capacity (CEC) of Na-MMT is 145 mmol/100 g. DDAC with a concentration of 70% was supplied by Shanghai 3D, Bio-chem Co., Ltd., Shanghai, China. The polyfluotetraethylene (PTFE) films were used as demoulding materials to avoid mould sticking to the board during hot-pressing.

### 2.2. Preparation of OMMT

Na-MMT were dispersed in distilled water at 5 wt % in a beaker and stirred for 2 h in a water bath at 60 °C. An allocated amount of DDAC equivalent to 0.7 CEC of Na-MMT was added into the Na-MMT suspension and reacted for another 2 h in the same condition. The remaining solution was centrifuged and the OMMT slurry was obtained. After washing seven times with distilled water to remove un-reacted DDAC, the OMMT slurry was dried at 120 °C in an oven, ground into a powder, and passed through a 200-mesh sieve.

### 2.3. Modification of PLA

Prior to processing, OMMT and PLA were dried at 103 °C for 24 h. OMMT at different contents and PLA powder were respectively mixed in a high-speed blender at about 2900 rpm for 4 min according to the formulation given in Table 1. After that, the mixture was extruded via a co-rotating twin-screw extruder (KESUN KS-20, Kunshan, China) with a screw diameter of 20 mm and a length-to-diameter ratio of 36/1. The corresponding temperature profile along the extruder barrel was 160/170/180/180/150 °C, and the screw speed was 180 rpm. The extrudate was cooled down in the air and ground into powder again. Afterward, the modified PLA was obtained.

### 2.4. Preparation of PLA/WF Composites

The composites contained 50 wt % WF and another 50 wt % virgin or OMMT-modified PLA. The oven-dried WF and PLA (at 103 °C for 24 h) powder were again mixed in the high-speed blender and extruded in the same manner as the process of PLA modification and broken into granules. A hot press (SYSMEN-II, China Academy of Forestry, Beijing, China) was used to produce the composites by compressing the mat at 180 °C with a pressure of 4 MPa for 6 min. About 300 g of granules were placed in a square mould with dimensions of 270 × 270 × 3 mm^3^. Prior to demoulding, the formed mat was cooled down at 4 MPa for another 6 min at room temperature. After then, all the mats were cut into required dimensions for further tests according to the related standards.

### 2.5. Characterization and Tests

X-ray diffraction (XRD) analysis of oven-dried Na-MMT and OMMT powders, virgin and OMMT-modified PLA, and the composites were carried out on an X-ray 6000 (Shimadzu, Japan) machine. The X-ray beam was Cu-Kα (λ = 0.154 nm) radiation, operated at 40 kV and 30 mA. The scanning rate was 2 °/s and 2θ ranged from 1.5° to 40°. By considering Bragg’s law, see Equation (1) [30], the interlayer distance of montmorillonite (MMT) can be calculated.
(1)$$d_{001}=\frac{n\lambda }{2\sin\theta }$$
where, *d*_001_ is the interlayer distance between (001) planes; n refers to the integer wavelength number (*n*=1); λ refers to the X-ray wavelength; *θ* refers to the maximum diffraction angle.

The distribution of OMMT in PLA and the composites was observed by TEM (JEOL 1010, Osaka, Japan) with an acceleration voltage of 100 kV. The samples were cut with an ultra-microtome diamond knife to obtain ultrathin (50 nm) sections. 

The water uptake (WU) and thickness swelling (TS) tests were carried out according to the Chinese standard GB/T 17657-2013 on the samples with a size of 50 × 50 × 3 mm^3^ with complete immersion in water at 20 ± 2 °C. The WU was calculated based on the weight percent gains at intervals of 6, 24, 48, 96, 144, 192, and 240 h. The weights were taken after the removal of excessive water on the sample surface. The TS was determined based on the mid-span thickness changes till the end of the test. The average WU, TS, and standard deviations were based on the values of six replicates. To further understand the water absorption process, the Fick’s diffusion coefficients of PLA and PLA composites were calculated according to Equation (2) [31].
(2)$$D=\frac{\pi }{16}{\left(\frac{h}{WU_{eq}}\right)}^{2}(\frac{\partial WU}{\partial \sqrt{t}}{)}^{2}$$
where, *D* is the diffusion coefficient, *WU_eq_* is the final equilibrium water uptake measured at the end of the test, $\frac{\partial WU}{\partial \sqrt{t}}$ is the slope taken from the water uptake WU vs. square root of time *t*, and *h* is the thickness of the sample.

The flexural test was carried out according to the Chinese standard GB/T 9341-2000, which involves a three-point bending test at a crosshead speed of 2 mm/min. The sample dimension is 60 × 25 × 3 mm^3^ with a gauge length of 50 mm. The tensile test was carried out according to the Chinese standard GB/T 1040-92 at a testing speed of 2 mm/min. The sample dimension is 250 × 25 × 3 mm^3^ with a gauge length of 100 mm. The Notched Charpy impact test was carried out according to the Chinese standard GB/T 1043.1-2008 with a size of 80 × 15 × 3 mm^3^. Twenty replicates were used in flexural, tensile, and impact tests.

The morphologies of virgin and OMMT-modified PLA and the composites were observed by SEM (Hitachi S-4800, Kyoto, Japan) with an acceleration voltage of 5 kV. All the samples were sputter-coated with gold prior to observation.

## 3. Results and Discussion

### 3.1. OMMT Characterization

The XRD patterns of Na-MMT and OMMT powders, virgin and OMMT-modified PLA, and the WF composites are shown in Figure 1. The crystal (001) plane diffraction peak of Na-MMT was located at 2θ = 6.05°, corresponding to the *d*_001_ value of 1.46 nm. After modifying with DDAC, the peak shifted to a lower location at 2θ = 3.98°, indicating the enlarged interlayer distance of MMT of 2.22 nm, see Figure 1a. This was because the long organic chain ion of DDAC was entered into the galleries through a cation exchange reaction. This characteristic peak can be also observed in OMMT-modified PLA samples, see Figure 1b, irrespective of OMMT contents. In addition, the interlayer distances were further enlarged to 3.68 nm (2θ = 2.40°) due to the infiltration of PLA chains into the silicate layers [28]. It should be noted that both the virgin and OMMT-modified PLA were almost amorphous because of the quick cooling [32]. The results could also be proved by the relatively low crystallinity of PLA, which was calculated using the XRD, see Table 1. However, the incorporation of OMMT could slightly improve the crystallinity of PLA due to the nucleating effect [25]. After the introduction of WF, the characteristic peak of WF at 2θ = 17° and 22.5° existed, while that of OMMT at 2θ = 2.40° in OMMT-modified PLA samples completely disappeared, see Figure 1c. Liao and Wu [33] added 7 wt % of clay into poly(ethylene-octene)/WF elastomer and also found no apparent peaks. They suggested that the clay had been fully exfoliated into individual and collapsed silicate layers, thus no regular crystalline structure could be obtained from the XRD patterns. As mentioned above, the WPC was prepared by first compounding OMMT with virgin PLA. Therefore, redistribution of OMMT may occur during the second extrusion process, forming a more uniform dispersion of the OMMT with exfoliated structures in the PLA matrix. However, the crystallinity was less affected by OMMT in the PLA/WF composites. More details could be provided by TEM images.

The TEM images of virgin and OMMT-modified PLA and the composites are shown in Figure 2. Due to the addition of large contents of discontinuous WF, the ultra-film of PLA/WF composites were easy to break, see Figure 2e–h. It can be seen that the surfaces of virgin PLA, see Figure 2a, and PLA/WF composites, see Figure 2e, appeared to be very clean. The cell wall of WF was clear in virgin PLA/WF. As for OMMT-modified samples, the OMMT was almost exfoliated in the 0.5/PLA sample, see Figure 2b, with considerable uniform dispersion and individual silicate layers. However, in 1.5/PLA and 2/PLA samples, see Figure 2c,d, most of the OMMT gathered into large particles, showing poor dispersion in PLA. In the 0.5/PLA/WF sample, see Figure 2f, the good dispersion and highly exfoliated structure of OMMT remained. It was interesting that in the 1.5/PLA/WF sample, see Figure 2g, the stacked OMMT redistributed evenly into PLA and formed exfoliated structures. The aggregates were present in a much lower quantity than in the 1.5/PLA sample. However, in 2/PLA/WF sample, see Figure 2h, aggregates were still present, which should be harmful to the properties of the composites. In the OMMT-modified PLA/WF composite samples, OMMT was not only located in the PLA matrix but also entered into the WF cell walls. Thus, it could be concluded that at low OMMT content (0.5 wt %), well-dispersed and highly exfoliated structures of OMMT existed in the PLA matrix. However, when the contents increased to 1.5 wt %, the aggregates appeared. The second blending process was beneficial to the redistribution and further exfoliation of OMMT, where the OMMT was not only dispersed in PLA matrix but also penetrated into the WF cell wall. However, at the highest OMMT content (2 wt %), aggregates still existed.

### 3.2. Water Uptake and Thickness Swelling

Figure 3 shows the WU values of virgin and OMMT-modified PLA and the composites. The WU values were found to increase with the increase of immersion time. The speed of WU increased fast initially and then slowed down.

Table 2 lists the final WU, diffusion coefficient, and thickness swelling of virgin and OMMT-modified PLA and the composites. The final WU values of virgin and OMMT-modified PLA were fairly low due to the hydrophobic nature of PLA. Thus, the thickness did not change after 240 h of water immersion. The small amount of absorbed water might account for the end hydroxyl groups in PLA. Incorporation of 0.5 and 1 wt % OMMT into PLA could slightly reduce the absorption of water, which may be because the highly exfoliated OMMT in PLA could act as a nucleating agent to improve the crystallinity of PLA. As the OMMT content increased, the nucleating effect was compensated by the agglomerated silicate layers. The addition of OMMT can also reduce the diffusion of water into PLA with decreased diffusion coefficient values because of the barrier effect of OMMT layers on water transport [34]. It can be seen that the diffusion coefficients of OMMT-modified PLA were lower than the virgin one, while the value was the lowest in 0.5/PLA and 1/PLA groups owing to the excellent dispersion. 

In the case of the PLA/WF composites, the values of water uptake, diffusion coefficient, and thickness swelling increased sharply because the WF easily absorbed water and was quick to hydro-expand. At a certain content of WF (50 wt %), the differences among these composites can be owed to different OMMT contents. Similar to the results of virgin and OMMT-modified PLA, the addition of OMMT lead to decreases of these values. Furthermore, OMMT penetrated into the WF cell wall can reduce accessible hydroxyl groups of WF, which consequently improved the compatibility with the non-polar PLA matrix, restricting both the water absorption and dimensional expansion. However, the optimal group was obtained at an OMMT content of 1.5 wt %, which was different to OMMT-modified PLA at a content of 0.5 or 1 wt %. As mentioned in the TEM analysis, the second preparation process contributed to the redistribution of OMMT in both PLA and WF. Therefore, the properties of composites could be further improved. 

### 3.3. Mechanical Properties

Table 3 lists the results of mechanical properties of virgin and OMMT-modified PLA and the composites. The mechanical properties were influenced by various parameters such as the original mechanical properties of the filler and fiber, fiber properties, and interfacial adhesion between the fiber and matrix [35]. The addition of WF into PLA had little effect on the flexural strength, but the tensile and impact strength decreased sharply due to the poor interfacial adhesion between WF and the PLA matrix. However, the flexural and Young’s modulus increased compared to virgin PLA. The increased modulus was associated with the increased stiffness and brittleness caused by WF for its high modulus. As for the virgin and OMMT-modified PLA samples, the 0.5/PLA sample had the largest flexural, tensile, and impact strength with values of 57.2 MPa, 43.9 MPa, and 10.9 J/m^2^, respectively. MMT was a type of material with high strength and hardness, which could absorb more of the stress and energy applied to the PLA [15]. Therefore, the highly exfoliated OMMT in the PLA matrix provided significant improvements. The biggest flexural and Young’s modulus were obtained in the 1.5/PLA sample because the modulus was more greatly associated with steric hindrance filler effects than with the interfacial bonding. The high content of OMMT should result in high stiffness. The decreases in the mechanical properties at 2 wt % OMMT content should be ascribed to the formation of an extremely weak interface caused by the agglomeration of OMMT. Considering the effect of the OMMT content on the mechanical properties of PLA/WF composites, the optimal group appeared to be at an OMMT content of 1.5 wt %, which should be owing to the redistribution of OMMT during the second extruding process as mentioned above. In our previous study [28], the OMMT-modified PLA/WF composites were prepared by first modifying WF with OMMT. The optimal group had an average flexural strength, flexural modulus, tensile strength, Young’s modulus, and impact strength at 75.5 MPa, 6.95 GPa, 56.9 MPa, 6.11 GPa, and 1.3 J/mm^2^, respectively. Except for the flexural modulus and impact strength, all values obtained in this study were lower, which should be associated with the weak interface. Thus, to further improve the properties of the composites, WF modification should be taken into account.

The SEM images of the fracture surfaces can give supporting information for the mechanical properties. Figure 4 shows the fracture surfaces of virgin and OMMT-modified PLA and the composites. The fracture surface of virgin PLA was very smooth, see Figure 4a, with some micro-holes. The addition of OMMT caused embrittlement of PLA. Folded surfaces were visible, see Figure 4b–d. At 2 wt % OMMT content, the surface was very rugged. Thus, the decreased mechanical properties were reasonable. As for the PLA/WF composites, big cracks were observed on the fracture surface of the virgin one, see Figure 4e, suggesting the debonding and pull-out of WF when force was applied. Incorporation of 0.5 and 1.5 wt % OMMT resulted in improved interfacial adhesion between WF and PLA, see Figure 4f,g. The WF was well-coated by PLA, suggesting the effectively transferred stress from the matrix to the WF. However, when the OMMT content went up to 2 wt %, see Figure 4h, cracks reappeared.

## 4. Conclusions

The modification of PLA with OMMT could improve the physical and mechanical properties of PLA and its resulting composites with WF. For PLA, the optimal sample of modified PLA was found at an OMMT content of 0.5 wt %, where good dispersion of OMMT was obtained with highly exfoliated structures. For PLA/WF composites, owing to the redistribution of OMMT during the second manufacturing process, the optimal group was found at an OMMT content of 1.5 wt %. The OMMT was not only dispersed into the PLA matrix but also penetrated into the WF cell wall. Thus, the interfacial adhesion between WF and PLA also improved. However, at the highest OMMT content (2 wt %), the physical and mechanical properties of PLA and the composites decreased due to the agglomeration of OMMT. 

## Figures and Tables

**Figure 1 polymers-11-00204-f001:**
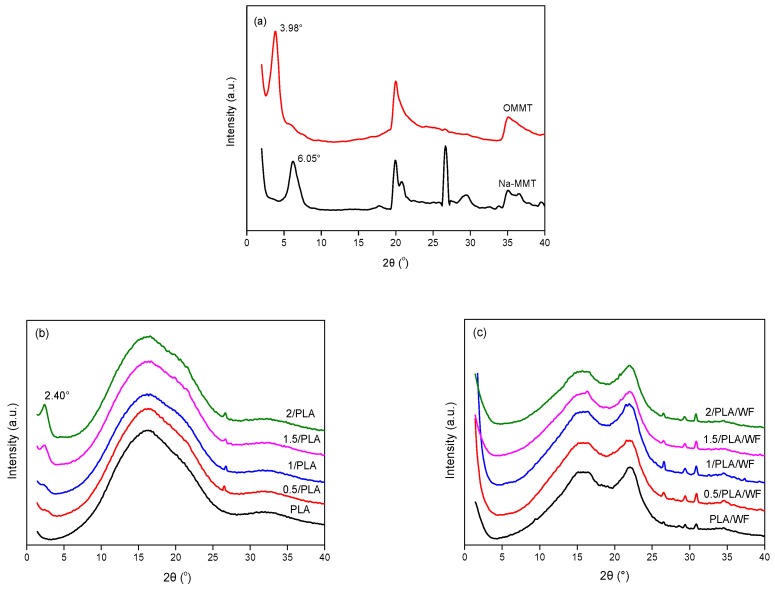
XRD patterns of Na-montmorillonite (MMT) and organo-montmorillonite (OMMT) powders (**a**), virgin and OMMT-modified poly(lactic acid) (PLA) (**b**), and the composites (**c**).

**Figure 2 polymers-11-00204-f002:**
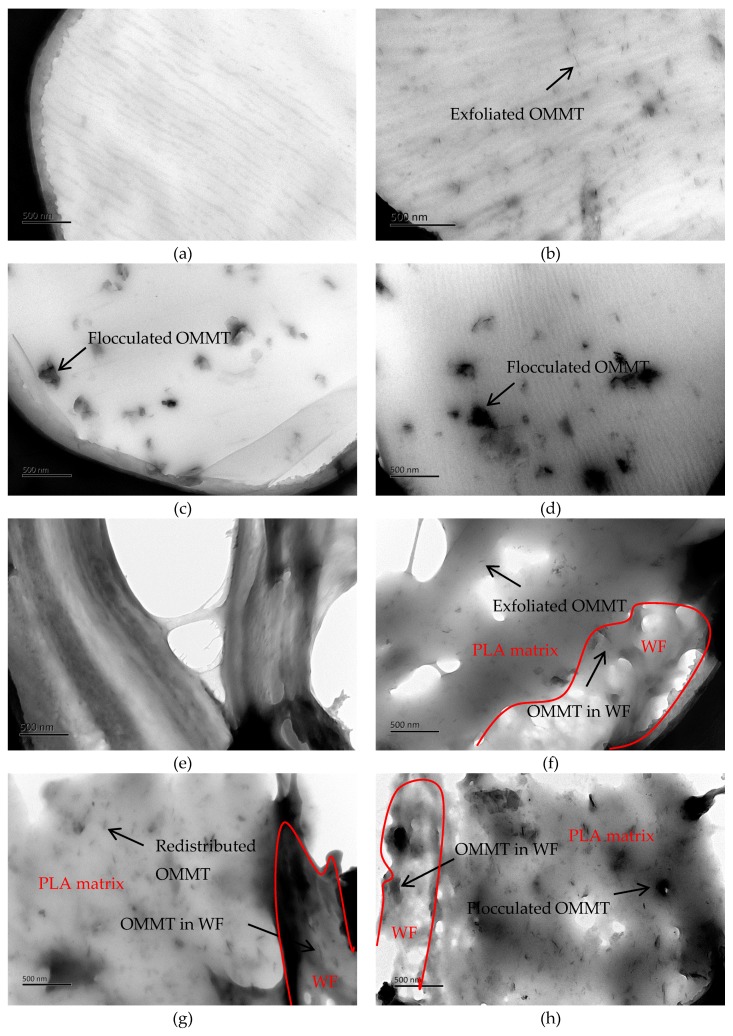
TEM images of virgin and OMMT-modified PLA and the composites. (**a**) PLA; (**b**) 0.5/PLA; (**c**) 1.5/PLA; (**d**) 2/PLA; (**e**) PLA/WF; (**f**) 0.5/PLA/WF; (**g**) 1.5/PLA/WF; (**h**) 2/PLA/WF.

**Figure 3 polymers-11-00204-f003:**
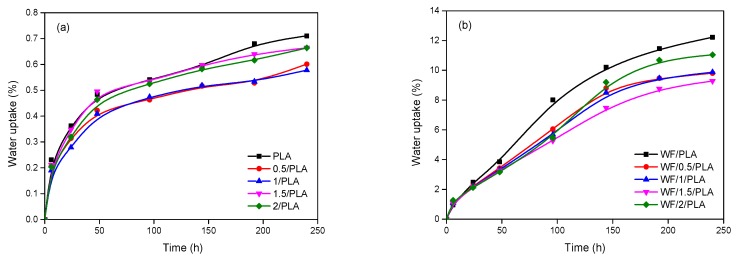
Water uptake of virgin and OMMT-modified PLA (**a**) and the composites (**b**).

**Figure 4 polymers-11-00204-f004:**
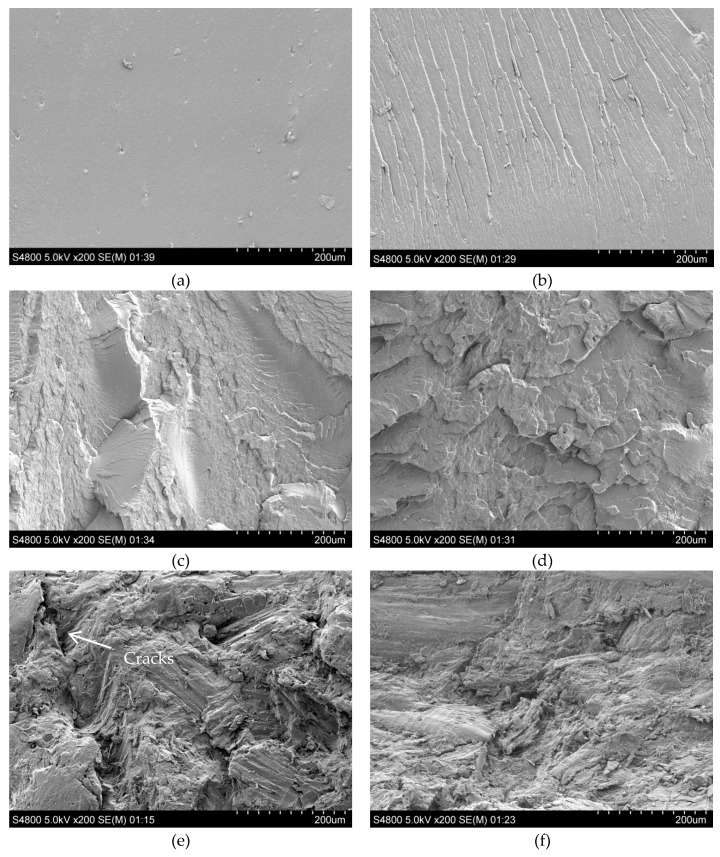
SEM micrographs of virgin and OMMT-modified PLA and the composites. (**a**) PLA; (**b**) 0.5/PLA; (**c**) 1.5/PLA; (**d**) 2/PLA; (**e**) PLA/WF; (**f**) 0.5/PLA/WF; (**g**) 1.5/PLA/WF; (**h**) 2/PLA/WF.

**Table 1 polymers-11-00204-t001:** Composition and labeling of the studied formulation.

Labels	OMMT (wt %)	PLA (wt %)	WF (wt %)
PLA	0	100	0
0.5/PLA	0.5	99.5	0
1/PLA	1	99	0
1.5/PLA	1.5	98.5	0
2/PLA	2	98	0
PLA/WF	0	50	50
0.5/PLA/WF	0.25	49.75	50
1/PLA/WF	0.5	49.5	50
1.5/PLA/WF	0.75	49.25	50
2/PLA/WF	1	49	50

**Table 2 polymers-11-00204-t002:** Final water uptake, diffusion coefficient, thickness swelling, and crystallinity of virgin and OMMT-modified PLA and the composites.

Labels	Final Water Uptake (%)	Diffusion Coefficient (×10^-10^) (m^2^/s)	Thickness Swelling (%)	Crystallinity (%)
PLA	0.71(0.01)	1.59(0.04)	0(0)	30.0
0.5/PLA	0.58(0.02)	1.38(0.02)	0(0)	31.5
1/PLA	0.60(0.03)	1.38(0.07)	0(0)	31.9
1.5/PLA	0.66(0.03)	1.46(0.03)	0(0)	32.3
2/PLA	0.66(0.05)	1.42(0.02)	0(0)	32.7
PLA/WF	12.22(0.12)	2.37(0.11)	12.68(0.58)	39.6
0.5/PLA/WF	9.81(0.08)	2.12(0.15)	9.26(0.34)	40.0
1/PLA/WF	9.87(0.20)	2.00(0.11)	8.79(0.17)	39.9
1.5/PLA/WF	9.28(0.10)	1.75(0.09)	7.36(0.26)	40.1
2/PLA/WF	11.04(0.27)	2.19(0.13)	10.68(0.64)	40.1

Note: Values in the parentheses represent the standard deviations of replicates.

**Table 3 polymers-11-00204-t003:** Mechanical properties of virgin and OMMT-modified PLA and the composites.

Labels	Flexural Strength (MPa)	Flexural Modulus (GPa)	Tensile Strength (MPa)	Young’s Modulus (GPa)	Impact Strength (J/m^2^)
PLA	33.5(2.4)	2.54(0.35)	29.0(1.8)	1.21(0.05)	3.4(0.4)
0.5/PLA	57.2(3.8)	3.08(0.40)	43.9(2.2)	1.29(0.09)	10.9(2.2)
1/PLA	42.2(4.5)	3.59(0.18)	37.3(1.0)	1.36(0.05)	7.3(0.6)
1.5/PLA	38.9(4.0)	4.72(0.17)	35.2(2.1)	2.13(0.18)	6.0(0.7)
2/PLA	38.1(2.0)	3.49(0.22)	29.1(0.9)	1.54(0.20)	4.8(0.7)
PLA/WF	38.0(1.2)	6.99(0.23)	18.7(2.9)	1.66(0.22)	2.2(0.7)
0.5/PLA/WF	40.9(5.4)	6.80(0.31)	20.9(3.9)	2.03(0.15)	3.6(0.9)
1/PLA/WF	48.3(4.3)	7.19(0.81)	28.5(1.7)	2.22(0.15)	3.9(0.5)
1.5/PLA/WF	55.4(5.9)	7.40(0.47)	32.9(2.1)	2.26(0.10)	5.5(0.4)
2/PLA/WF	38.2(3.3)	7.33(0.44)	25.1(3.0)	2.03(0.19)	3.6(0.4)

Note: Values in the parentheses represent the standard deviations of replicates.

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
