# Peer review of "Preparation of Organo-Montmorillonite Modified Poly(lactic acid) and Properties of Its Blends with Wood Flour"

_polymers, 2019, doi:10.3390/polym11020204_

Round 1
Reviewer 1 Report
The manuscript entitled “Preparation of organo-montmorillonite modified poly (lactic acid) and
properties of its blends with wood flour” by Ru Liu et al. covers important problem of development of biodegradable hybrid polymeric composites filled with different filler types. Detailed description of composites properties changes induced by incorporation of the OMMT and wood flour was prepared comprehensively. Article is well written, moreover results described in submitted work may be useful for further investigations and became a valuable source information of academic and industrial investigations. As a whole the topic addressed by the authors is relatively new and within the scope of the Journal but before the paper can be accepted for publication, some issues need to be addressed.
- (Line 14 ) I recommend to change to poly(lactic acid) (PLA) / wood flour composites (WF) because PLA is the matrix.
- (Line21 and 279) 2 wt% is not extremely high OMMT content, rather the highest filler content used in this study.
- (Line 81) The MFI range is really high (10-20). Is it possible that polymeric material is characterized with non-uniform structure? Please comment this fact.
- (Line 100) Please provide information about cooling medium used after extrusion (water or air).
- (Preparation of the samples) How and how long materials were dried before processing? Unmodified wood-flour as well as MMT reveal high ability of moisture absorption. Did authors realize investigations that confirm lack of hydrolytic degradation during melt processing caused by presence of residual moisture contained in fillers? Realization of even simple MFI test may be useful in this case.
- (Line 112) Authors should use the same order of presented methods in "2.5" section as in "Results and discussion" section.
- (Line 134) Preparation of impact test of the composite materials with using only 8 specimens may raise doubts, because of high influence on structural defects in composite structure on this mechanical parameter and its deviations. I recommend to complement test with higher amount of the specimens.
- (Line 181) Authors write about penetration of PLA into WF cell wall, while in Figures this phenomenon truly cannot be seen clearly. While Authors for sure know how images were prepared and they seen all details, reader may be confused. Please include at TEM images (f-h) additional sign to present interface between WF and polymeric matrix and/or presence of WF cell.
- (Line 204) Authors suggest that differences in water absorption between single-filler composite samples may be an effect of OMMT nucleating ability on PLA, while XRD analysis did not confirm this fact. Moreover, authors write that PLA-OMMT composites were characterized with almost amorphous structure. I would ask to comment this or evaluate the influence of the fillers on PLA crystallinity.
- Authors many time discuss the differences between PLA-OMMT and PLA-OMMT-WF composites in a course of different dispersion of the filler caused by different thermo-mechanical history and in fact different processing method. In order to fully understand the changes in a structure and properties, both composites should be prepared in the same manner, so the PLA-OMMT composites also should be melt processed twice in order to exclude speculations. I understand that new series of the composites will not be prepared, however in future work I recommend to evaluate the samples with the same processing history.
Author Response
We must thank you for the valuable comments and suggestions, which helped improve our manuscript greatly. Please do forward our heartfelt thanks to the reviewers. Based on the comments we received, careful modifications have been made to the manuscript. All changes were marked in red text. We hope that the revised manuscript answered the questions. Below you will find our point-by-point responses to the comments/ questions:
To Reviewer 1:
The manuscript entitled “Preparation of organo-montmorillonite modified poly (lactic acid) and properties of its blends with wood flour” by Ru Liu et al. covers important problem of development of biodegradable hybrid polymeric composites filled with different filler types. Detailed description of composites properties changes induced by incorporation of the OMMT and wood flour was prepared comprehensively. Article is well written, moreover results described in submitted work may be useful for further investigations and became a valuable source information of academic and industrial investigations. As a whole the topic addressed by the authors is relatively new and within the scope of the Journal but before the paper can be accepted for publication, some issues need to be addressed.
- (Line 14) I recommend to change to poly(lactic acid) (PLA) / wood flour composites (WF) because PLA is the matrix.
We have changed it to PLA/WF composites, and also corrected it throughout the manuscript.
- (Line21 and 279) 2 wt% is not extremely high OMMT content, rather the highest filler content used in this study.
We have corrected the description to “the highest OMMT content”.
- (Line 81) The MFI range is really high (10-20). Is it possible that polymeric material is characterized with non-uniform structure? Please comment this fact.
The original MFI value was obtained from the company. To narrow the range, we retested the MFI for the PLA and it was 12-15 g/10 min with the load of 2.16 kg. We have corrected in the manuscript in page 2 line 84. “It has a density around 1.25 g/cm3 and a melt flow index about 12-15 g/10min at 190oC with the load of 2.16kg.”
- (Line 100) Please provide information about cooling medium used after extrusion (water or air).
The cooling medium used after extrusion was in the air. We have added it in the manuscript in page 3 line 118. “The extrudate was cooled down in the air”. Please check.
- (Preparation of the samples) How and how long materials were dried before processing? Unmodified wood-flour as well as MMT reveal high ability of moisture absorption. Did authors realize investigations that confirm lack of hydrolytic degradation during melt processing caused by presence of residual moisture contained in fillers? Realization of even simple MFI test may be useful in this case.
The raw materials before processing were dried in an oven at 103 oC for 24h. We have added it in the manuscript in page 3 line 113. “Prior to processing, the OMMT and PLA were dried at 103 oC for 24 h.”
It was known that the WF underwent some thermal degradation during both the extruding and hot-pressing process for about 2-5 wt% mass loss, to minimize the effects of moisture content on the thermal degradation process, the WF, OMMT and the PLA were oven dried prior to the manufacturing.
After adding WF, the MFI of the composites was hard to be obtained by the same method of testing neat PLA due to the significant increase of resistance.
- (Line 112) Authors should use the same order of presented methods in "2.5" section as in "Results and discussion" section.
We have corrected the order of the “2.5” section.
- (Line 134) Preparation of impact test of the composite materials with using only 8 specimens may raise doubts, because of high influence on structural defects in composite structure on this mechanical parameter and its deviations. I recommend to complement test with higher amount of the specimens.
We have retested all the mechanical tests for twenty replicates, and the data were updated in Table 3.
- (Line 181) Authors write about penetration of PLA into WF cell wall, while in Figures this phenomenon truly cannot be seen clearly. While Authors for sure know how images were prepared and they seen all details, reader may be confused. Please include at TEM images (f-h) additional sign to present interface between WF and polymeric matrix and/or presence of WF cell.
We have added signs for the TEM images for the WF and PLA matrix as well as the interface.
- (Line 204) Authors suggest that differences in water absorption between single-filler composite samples may be an effect of OMMT nucleating ability on PLA, while XRD analysis did not confirm this fact. Moreover, authors write that PLA-OMMT composites were characterized with almost amorphous structure. I would ask to comment this or evaluate the influence of the fillers on PLA crystallinity.
We added the crystallinity results in Table 2. It can be seen that the OMMT slightly improved the crystallinity of PLA. As a result, it caused little decrease of final WU value. Some discussion was also added in page 4 line 180. “The results could be also proved by the relative low crystallnity of PLA, which was calculated by the XRD (Table 1). However, the incorporation of OMMT could slightly improve the crystallinity of PLA due to the nucleating effect [25]..” And page 4 line 190. “However, the crystallinity was less affected by OMMT in the PLA/WF composites.”
- Authors many time discuss the differences between PLA-OMMT and PLA-OMMT-WF composites in a course of different dispersion of the filler caused by different thermo-mechanical history and in fact different processing method. In order to fully understand the changes in a structure and properties, both composites should be prepared in the same manner, so the PLA-OMMT composites also should be melt processed twice in order to exclude speculations. I understand that new series of the composites will not be prepared, however in future work I recommend to evaluate the samples with the same processing history.
Thanks for your sincere suggestions. We will take it and evaluate the influence in future study.
Sincerely yours,
Ru Liu, Xiaoqian Yin, Anmin Huang, Chen Wang, and Erni Ma
Jan 12, 2019

Reviewer 2 Report
The authors did a good job presenting their research in an organized manner. The findings and conclusions are reasonable. There are some grammatical errors which should be addressed throughout the paper.
Author Response
We must thank you for the valuable comments and suggestions, which helped improve our manuscript greatly. Please do forward our heartfelt thanks to the reviewers. Based on the comments we received, careful modifications have been made to the manuscript. All changes were marked in red text. We hope that the revised manuscript answered the questions. Below you will find our point-by-point responses to the comments/ questions:
To Reviewer 2:
The authors did a good job presenting their research in an organized manner. The findings and conclusions are reasonable. There are some grammatical errors which should be addressed throughout the paper.
We have corrected some grammatical errors throughout the manuscript. Please check.
Sincerely yours,
Ru Liu, Xiaoqian Yin, Anmin Huang, Chen Wang, and Erni Ma
Jan 12, 2019

Reviewer 3 Report
I reviewed the manuscript entitled “Preparation of organo-montmorillonite modified poly (lactic acid) and properties of its blends with wood flour”. The authors proposed a wood/PLA composite, in which the organo-montmorillonite was introduced to enhance the composite performance. The authors well presented the scientific concepts. All the materials come from nature, which promote this research interests to other researchers. I suggest a minor revision before being published. Detail comments as follows:
1) The authors may include more articles about the composites made from polymer-matrix nanoparticles and natural fibers.
2) Line 129-134. The authors only mentioned the sample sizes for tensile and flexural tests, however, no gauge lengths were described.
3) Line 140. The Eq. (1) might be moved to Materials and Methods section.
4) Fig. 1 and Fig. 3. The background (including color and dots) of these two figures should be wipe away.
5) Line 196. Eq (2) belongs to method description, please move to Materials and Methods section.
6) Line 272. I believe “Discussion” should be “Conclusions”.
Author Response
We must thank you for the valuable comments and suggestions, which helped improve our manuscript greatly. Please do forward our heartfelt thanks to the reviewers. Based on the comments we received, careful modifications have been made to the manuscript. All changes were marked in red text. We hope that the revised manuscript answered the questions. Below you will find our point-by-point responses to the comments/ questions:
To Reviewer 3:
I reviewed the manuscript entitled “Preparation of organo-montmorillonite modified poly (lactic acid) and properties of its blends with wood flour”. The authors proposed a wood/PLA composite, in which the organo-montmorillonite was introduced to enhance the composite performance. The authors well presented the scientific concepts. All the materials come from nature, which promote this research interests to other researchers. I suggest a minor revision before being published. Detail comments as follows:
1) The authors may include more articles about the composites made from polymer-matrix nanoparticles and natural fibers.
We added two articles of reference 15 and 17 about the polymer nanoparticles and natural fibers.
15. Saba, N.; Md Tahir, P.; Jawaid, M. A review on potentiality of nano filler/natural fiber filled polymer hybrid composites. Polymers 2014, 6, 2247-2273.
17. Zhou, H.; Hao, X.; Wang, H.; Wang, X.; Liu, T.; Xie, Y.; Wang, Q. The reinforcement efficacy of nano- and microscale silica for extruded wood flour/HDPE composites: the effects of dispersion patterns and interfacial modification. J. Mater. Sci. 2018, 53, 1899-1910.
2) Line 129-134. The authors only mentioned the sample sizes for tensile and flexural tests, however, no gauge lengths were described.
The gauge lengths of flexural and tensile tests were 50 mm and 100 mm, respectively. We added it in the manuscript.
3) Line 140. The Eq. (1) might be moved to Materials and Methods section.
We have moved Eq.(1) to the Materials and Methods section.
4) Fig. 1 and Fig. 3. The background (including color and dots) of these two figures should be wipe away.
We have corrected Fig.1 and Fig.3.
5) Line 196. Eq (2) belongs to method description, please move to Materials and Methods section.
We have moved Eq.(2) to the Materials and Methods section.
6) Line 272. I believe “Discussion” should be “Conclusions”.
We have corrected this mistake.
Sincerely yours,
Ru Liu, Xiaoqian Yin, Anmin Huang, Chen Wang, and Erni Ma
Jan 12, 2019

Round 2
Reviewer 1 Report
Thank you for your answers and including suitbale corrections in revised version of the manuscript.